# Preparation of Aminated Sodium Lignosulfonate and Efficient Adsorption of Methyl Blue Dye

**DOI:** 10.3390/ma17051046

**Published:** 2024-02-24

**Authors:** Li-Zhu Huo, Chao-Fei Guo, Zhu-Xiang Gong, Hao Xu, Xue-Juan Yang, Yu-Xuan Wang, Xi-Ping Luo

**Affiliations:** 1College of Chemistry and Materials Engineering, Zhejiang A&F University, Hangzhou 311300, China; hlz@stu.zafu.edu.cn (L.-Z.H.); chaoguo@zafu.edu.cn (C.-F.G.); 2022005@stu.zafu.edu.cn (Z.-X.G.); jasonxu@stu.zafu.edu.cn (H.X.); yangxj@zafu.edu.cn (X.-J.Y.); 2Zhejiang Provincial Key Laboratory of Chemical Utilization of Forestry Biomass, Hangzhou 311300, China

**Keywords:** sodium lignosulfonate, amination, adsorption, methyl blue dye

## Abstract

The aminated sodium lignosulfonate (AELS) was prepared through a Mannich reaction and characterized via FT-IR, TG, SEM and XPS in this study. Subsequently, the adsorption capacity of AELS for methyl blue (MB) was evaluated under various conditions such as pH, adsorbent dosage, contact time, initial concentration and temperature. The adsorption kinetics, isotherms and thermodynamics of AELS for methyl blue were investigated and analyzed. The results were found to closely adhere to the pseudo-second-order kinetic model and Langmuir isotherm model, suggesting a single-molecular-layer adsorption process. Notably, the maximum adsorption capacity of AELS for methyl blue (153.42 mg g^−1^) was achieved under the specified conditions (*T* = 298 K, *M_AELS_* = 0.01 g, pH = 6, *V_MB_* = 25 mL, *C*_0_ = 300 mg L^−1^). The adsorption process was determined to be spontaneous and endothermic. Following five adsorption cycles, the adsorption capacity exhibited a minimal reduction from 118.99 mg g^−1^ to 114.33 mg g^−1^, indicating good stability. This study contributes to the advancement of utilizing natural resources effectively and sustainably.

## 1. Introduction

During the past few decades, dyes have been widely used in various industries, including textiles, papermaking and printing [1]. However, water pollution caused by dyes has become increasingly serious with the rapid development of dyeing industries. Dyes impart serious threats to the environment, soil, aquatic ecosystems and human health due to their high toxicity, refractory to degradation and environmental persistence. The introduction of approximately 21 tons of toxic dyes into the water cycle each year worsens the already severe pollution problem [2,3,4].

Methyl blue (MB) is an aromatic heterocyclic anionic dye commonly used for staining live protozoa, bacteria and nerve cells in animal tissues. The chemical formula of methyl blue is shown in Appendix A. However, it is toxic and bioaccumulative to both organisms and the environment and can induce mutagenic effects [5]. Therefore, the content of dyes should be eliminated or minimized as much as possible ahead of the dye wastewater being discharged into water bodies. Now, many techniques have been developed and widely applied to solve such water pollution caused by dyes, including membrane separation, chemical oxidation, biodegradation, photocatalysis and adsorption [6,7,8,9,10]. Adsorption turns out to be one of the most appealing methods among them due to its rather easy operation, low cost, straightforward design and lack of secondary contamination [11,12]. This process transfers the dye from the effluent of water to the solid phase, thereby minimizing the volume of the effluent [13]. Many materials have been reported for the adsorption of methyl blue, such as Mn/MCM-41, KGM/GO hydrogel, Organo-bentonite, etc. [14,15,16]. However, these materials are non-renewable and have certain environmental hazards. Because lignin is inexpensive, readily available, biodegradable and infrequently utilized as a methyl blue adsorbent, there has been growing attention around it.

In the kingdom of plants, lignin is the second largest biomass resource after cellulose, accounting for about 15–30% of plant cell walls [17,18]. It is estimated that up to 150 billion tons of lignin are produced annually worldwide from plant growth. Lignin is an aromatic three-dimensional polymer composed of three types of phenylpropane unit types, guaiacyl, syringyl and para-hydroxyl, connected by different ether bonds and carbon–carbon bonds [19,20]. Numerous functional groups, including phenolic hydroxyl, alcohol hydroxyl, carbonyl, carboxyl, methoxy and conjugated double bonds, are present in it [21]. Sodium lignosulfonate (ELS) is similar to lignin in structure and is a water-soluble sulfonated lignin derivative, with a unique three-dimensional network structure. ELS is highly soluble in water; hence, as the adsorbent, it has a limited adsorption capability. Once modified, it can be used in a variety of fields. Of those, the Mannich reaction is the most straightforward and convenient. In acidic environments, amine groups are positively charged and ionizable [22].

In this study, a low-cost, recyclable and eco-friendly aminated sodium lignosulfonate (AELS) was prepared through the Mannich reaction. Orthogonal experiments were designed and carried out to compare the effects of reaction temperature, reaction time and diethylenetriamine (DETA) and formaldehyde (FA) quantities on the synthesis. FT-IR, TG, SEM and XPS were used to characterize the structure and morphology of AELS. The influences on methyl blue adsorption of pH, adsorbent dosage, contact time, initial concentration and temperature were investigated. The adsorption kinetics, isotherms, thermodynamics and the regeneration of methyl blue on AELS were also discussed. By utilizing the biomass resource lignin as an adsorbent for methyl blue, this study contributes to the sustainable development of natural resources and aligns with the principles of the circular economy. Furthermore, this research expands the potential applications of lignin, being beneficial for solving the problems of dye wastewater and resource utilization.

## 2. Materials and Methods

### 2.1. Materials

Sodium lignosulfonate (C_20_H_24_Na_2_O_10_S_2_) was purchased from Aladdin (Shanghai, China); 36~38 wt.% formaldehyde aqueous solution (HCHO) was purchased from Energy Chemical (Shanghai, China). Diethylenetriamine (C_4_H_13_N_3_) is an analytical-grade reagent (AR), and methyl blue (C_37_H_27_N_3_Na_2_O_9_S_3_) is a biological stain (BS); these were purchased from Macklin (Shanghai, China). Isopropanol (C_3_H_8_O), sodium hydroxide (NaOH) and hydrochloric acid (HCl) are AR, purchased from Sinopharm Chemical Reagent Co., Ltd. (Shanghai, China). Deionized water was obtained from a water purification system and was used to prepare samples and solutions as needed.

### 2.2. Synthesis of AELS

Using ultrasound, a 250 mL flask with three necks was filled with a homogenous solution composed of 20 mL of 0.5 mol L^−1^ sodium hydroxide solution and 5 g of ELS. The aforementioned combination was then heated to a specific temperature, and DETA was added. Following a thorough mixing of the aforementioned solution, a specific amount of FA was gradually added while magnetic stirring and condensation reflux were in place. A schematic representation of the synthesis of AELS is shown in Figure 1. After the reaction was over, the AELS solution was adjusted to acidic with 1 mol L^−1^ of hydrochloric acid until no brown precipitate precipitated. The precipitate was dried under vacuum at 50 °C and then ground to obtain AELS after being repeatedly washed with isopropanol and petroleum ether as well as deionized water until the washing liquid was neutral. Then, orthogonal experiments were conducted to determine the optimal conditions for the synthesis of AELS.

### 2.3. Characterization

The functional groups of the samples were characterized with a Fourier transform infrared spectrometer (FT-IR, Thermo Nicolet 6700, Waltham, MA, USA), and the samples were obtained using the potassium bromide tablet method. The test frequency range was 4000 cm^−1^~500 cm^−1^. The thermal stability of the samples was characterized through thermogravimetric analysis (TG, Netzsch TG 209 F3 Tarsus, Selb, Germany). The experimental temperature range was 30~800 °C, and the heating rate was 10 °C min^−1^. The morphology of the samples was characterized with field emission scanning electron microscopy (SEM, ZEISS Sigma 300, Oberkochen, Germany). The chemical composition and elements of the samples were characterized with X-ray photoelectron spectroscopy (XPS, Thermo Scientific K-Alpha, Waltham, MA, USA). A Zeta potential analyzer (Zeta Plus; S/N: 220010, Spring House, PA, USA) was used to measure the surface charge distributions of AELS at various pH values. Deionized water was mixed with AELS to create 0.1 wt.% dispersion solutions. To modify the pH of the mixtures to various values, 0.1 mol L^−1^ HCl and NaOH solutions were then used. A suitable volume of dispersion solutions was used to measure the zeta potential at room temperature. Every experiment was run three times, and the mean result was used.

### 2.4. Adsorption Experiment

The static adsorption method was applied to batchwise adsorption experiments. A batch of 25 mL methyl blue solutions at various concentrations was mixed with 0.01 g of AELS. Investigations were conducted on the effects of pH value, adsorbent dosage, initial concentration, contact time and temperature. To adjust the original pH of the solutions, 0.1 mol L^−1^ HCl and NaOH solutions were used. The colorimetric tube was stirred on an orbital shaker for a desired time at a constant working temperature of 298 K. We performed thermodynamic experiments at three distinct temperatures: 298 K, 308 K and 318 K. Following the adsorption time, the supernatant underwent a 5 min, 8000 r min^−1^ centrifugation. Using deionized water as a reference, a UV spectrophotometer operating at the dye’s maximum absorption wavelength of λ = 600 nm was used to measure the concentration of methyl blue in the supernatant solution both before and after the adsorption. Only the averages from each triplicate adsorption experiment were submitted in advance of this study. According to the standard curve generated for the methyl blue dye (Appendix A), the adsorption capacity *q_e_* (mg g^−1^), *q_t_* (mg g^−1^) and removal rate *R* (%) of the dye under different conditions were calculated according to Formulas (1)–(3):(1)$$q_{e}=\frac{C_{0}-C_{e}}{M}\times V$$
(2)$$q_{t}=\frac{C_{0}-C_{t}}{M}\times V$$
(3)$$R=\frac{C_{0}-C_{e}}{C_{0}}\times 100\%$$

In the formulas, *q_e_* is the equilibrium adsorption capacity (mg g^−1^); *q_t_* is the adsorption capacity at time *t* (mg g^−1^); *R* is the removal rate (%); *C*_0_, *C_e_* and *C_t_* are, respectively, the initial, residual, and equilibrium concentrations of the methyl blue dye (mg L^−1^); *V* is the volume of the dye (L); and *M* is the dosage of the adsorbent (g).

## 3. Results and Discussion

### 3.1. Optimization of the Synthetic Conditions of AELS

The study employed orthogonal array experiments based on L16(4^4^) to evaluate how various synthetic parameters influence the adsorption capacities of AELS [23]. The effects of the reaction temperature (A), reaction time (B), DETA quantity (C) and FA quantity (D) were investigated, and their levels are shown in Appendix A. The results of the orthogonal experiments are shown in Appendix A. In order to assess the impact of the different synthetic conditions and determine the optimal parameters for the synthesis of AELS, a range analysis was conducted on the removal rate (%) of the 16 synthesized AELS for methyl blue. The study focused on three vital parameters, K_ij_, k_ij_ and R_i_, each of which plays a significant role in assessing the effectiveness of the synthetic conditions. Here, i represents the factors (i = A, B, C, D), and j denotes the levels (j = 1, 2, 3, 4). Specifically, K_ij_ represents the cumulative removal rate at level j for factor i, the mean value of Kij is denoted as k_ij_, and Ri signifies the range between the maximum and minimum k_ij_ values across the four levels examined. To provide a clearer illustration of the impact of synthetic conditions, the average removal rate values, k_ij_, under different conditions are graphically represented in Figure 2. As can be seen, the degrees of impact for these four factors (reaction temperature, reaction time and the quantities of DETA and FA) in the synthesis reaction are as follows: DETA quantity > FA quantity > reaction temperature > reaction time. Under alkaline conditions, ELS lost a proton, transforming into a carbocation. DETA and FA undergo an addition reaction, and the resulting addition product reacted with the activated ELS through a substitution reaction, producing the amine-containing compound AELS [24]. Accordingly, the optimal levels of the various factors were A_4_, B_3_, C_1_, and D_4_, namely, a temperature of 85 °C, reaction time of 4 h, 3 g of DETA, and 4.5 mL of FA. All of the AELSs used in the characterization tests and adsorption experiments that followed were synthesized at the optimal level.

### 3.2. Characterization of AELS

The structures of ELS and AELS were characterized by FT-IR spectroscopy, as shown in Figure 3. Following the modification of the ELS, new vibration absorption peaks emerged, indicating changes in the functional groups on the surface. Most of the original functional groups were retained, as illustrated in the figure, with additional vibration peaks appearing at various positions. The peak at 3440 cm^−1^ was associated with the stretching vibration of -OH. Furthermore, the peaks at 2940 cm^−1^ were attributed to the asymmetric stretching vibrations of -CH_3_ and -CH_2_, while those at 2860 cm^−1^ were related to the symmetric stretching vibrations of -CH_3_ and -CH_2_ [25,26]. The peak at 1600 cm^−1^ was indicative of the aromatic skeleton vibration of lignin, whereas the presence of the characteristic peak of -SO_3_H at 1040 cm^−1^ also highlighted the original functional groups. Subsequent to the amination modification, fresh vibration absorption peaks manifested: the peak at 1630 cm^−1^ represented the in-plane bending vibration of N-H in NH_2_, the peak at 1350 cm^−1^ denoted the C-N stretching vibration in C-NH_2_, and the peak at 1090 cm^−1^ signified the C-N stretching vibration [27]. Overall, the results of the FT-IR analysis confirmed the successful grafting of amine groups onto ELS.

The TG and DTG curves of ELS and AELS are exhibited in Figure 4a,b. Both ELS and AELS exhibited a similar trend in their thermogravimetric curves. The weight loss process of both ELS and AELS could be distinguished into three distinct stages. The initial stage of weight loss primarily involved the evaporation of moisture from the samples. The second stage was the cleavage of aryl ethers, which was accompanied by deoxidation and decarboxylation reactions. The third stage was the cleavage of the C-C bonds of lignin structural units and the benzene ring fatty side chains. The DTG peaks of ELS and AELS were 266 °C and 349 °C, respectively. ELS exhibited a temperature of 166 °C at a 5% mass loss, while AELS had a temperature of 88 °C at the same mass loss. Similarly, when reaching a 50% mass loss, ELS demonstrated a temperature of 677 °C compared to AELS’s temperature of 480 °C. And at 800 °C, the residual mass of AELS was less than that of ELS. The thermal stability of AELS was not equivalent to that of ELS.

Figure 5 shows SEM and EDS images of ELS and AELS. As can be seen from the figure, at the same magnification scale (2 µm), ELS had spherical and irregular structures, and AELS had blocky porous structures. Compared with ELS, the porous surface of AELS was more conducive to the adsorption of dyes. The C element increased while the O element decreased, and the N element increased from 0.45% to 7.42%, indicating that the AELS contained amine groups.

XPS was used to investigate the elemental distributions of the samples in further detail. As demonstrated in Figure 6a, two peaks at 284 and 531 eV were observed in the XPS profiles of ELS and AELS, which correspond to C 1 s and O 1 s, respectively [28]. Comparing the full spectra of ELS and AELS, the N 1 s peak intensity of AELS increased, which demonstrates the introduction of amine groups. The ratios of C and O had increased, due to the introduction of amine groups with an alkyl group on the carbon chain of lignin, resulting in a higher carbon content and a lower oxygen content, consistent with the EDS results [29]. As shown in Figure 6b, a new -NH peak appeared at 401 eV accompanied by a rise in -NH_2_ intensity at 399 eV. In contrast to ELS, there were four peaks in the C 1 s spectrum and two peaks in the O 1 s spectrum of AELS. As can be seen from Figure 6c, ELS had three spectral peaks in 282~290 eV, namely C-C, C-OH and C-O-C/C=O, while AELS had a new peak of C-N at 285 eV, further confirming the introduction of amine groups in AELS. In addition, in the O 1 s high-resolution spectra, the C-O peak appeared at 531 eV (Figure 6d). These changes in the N, C and O spectra indicate that the amine groups were successfully grafted onto ELS.

### 3.3. Results of Adsorption Experiment

#### 3.3.1. The Effect of pH on the Adsorption Capacity

The pH significantly impacts adsorption processes by influencing the formation of functional groups in reactants and affecting the interaction between adsorbents and adsorbates [30,31]. To assess how pH affects the adsorption capacity of AELS, experiments were carried out at *C*_0_ = 50 mg L^−1^, *V* = 25 mL, *M* = 0.01 g, a pH range of 3~10 and *T* = 298 K. As can be seen from Figure 7a, after the pH of the solution was changed from 3 to 10, the adsorption capacity of AELS decreased from 119.46 mg g^−1^ to 5.59 mg g^−1^. At acidic pH, AELS was protonated to create amine groups, and the surface activation was given a positive charge. Subsequently, it enhanced the methyl blue molecules’ electrostatic attraction to the AELS surface [32]. According to Figure 7b, AELS had a positive charge when its pH was less than 7. As a result, preferential adsorption would result from a strong electrostatic interaction between positively charged AELS and negative charges in methyl blue. In addition, the -OH group on the surface of AELS formed hydrogen bonds with the N-H groups in the methyl blue molecules, which further promoted adsorption and then achieved adsorption equilibrium. At basic pH, methyl blue’s ability to adsorb would be inhibited because more OH^−^ would engage in competitive adsorption at a higher pH value. Additionally, ion exchange might occur during the adsorption process [33]. The degree of the protonation of amino groups reduced as pH increased, and the ionization of functional groups on AELS that contain oxygen, including carboxyl and hydroxyl groups, was encouraged. This results in a shift in the overall charge of AELS from positive to negative, with the emergence of an isoelectric point around pH 6.5. Therefore, the adsorption capacity decreased as pH increased. Notably, the difference in adsorption capacities between pH = 3 (119.46 mg g^−1^) and pH = 6 (118.99 mg g^−1^) was minimal, prompting the selection of pH = 6 as the optimal pH for the subsequent investigation.

#### 3.3.2. The Effect of Adsorbent Dosage on the Adsorption Capacity and Removal Rate

Experiments were conducted at *C*_0_ = 50 mg L^−1^, *V* = 25 mL, pH = 6, an *M* range of 0.005~0.05 g and *T* = 298 K to evaluate the impact of adsorbent dosage on both the adsorption capacity and removal rate. Figure 8 shows the relationship between adsorbent dosage, adsorption capacity and removal rate. As the amount of adsorbent dosage was increased from 0.005 g to 0.05 g, it was observed that the adsorption capacity decreased gradually from 132.21 mg g^−1^ to 24.82 mg g^−1^, while the removal rate increased from 52.88% to 99.28%. Both the overall surface area and the number of surface-active sites increased as the adsorbent dosage increased, thus improving the potential for hydrogen bonding. Nonetheless, the amount of dye molecules in the solution remained constant, causing some active sites to not reach full adsorption. Therefore, when the dosage of the adsorbent rose, the adsorption capacity declined, and the removal rate increased. Consequently, the best dosage for the remainder of the investigation was determined to be 0.01 g of adsorbent.

#### 3.3.3. Adsorption Kinetics

In this study, experiments were conducted in order to analyze the adsorption process and determine the underlying mechanism. The experimental conditions included *C*_0_ = 50 mg L^−1^, *V* = 25 mL, pH = 6, *M* = 0.01 g and *T* = 298 K. Three different kinetic models were utilized to fit the experimental data: the pseudo-first-order kinetic model, pseudo-second-order kinetic model, and intra-particle diffusion model. Equations (4)–(6), representing these kinetic models, were employed to analyze the adsorption process [34,35,36]:(4)$$q_{t}=q_{e}(1-e^{-k_{1}t})$$
(5)$$q_{t}=\frac{{q_{e}}^{2}k_{2}t}{1+q_{e}k_{2}t}$$
(6)$$q_{t}=k_{p}t^{1/2}+C$$In the formulas, *k*_1_ is the pseudo-first-order kinetic constant (min^−1^), *k*_2_ is the pseudo-second-order kinetic constant (g mg^−1^ min^−1^), *k_p_* is the intra-particle diffusion kinetic constant (mg g^−1^ min^−1/2^), and *C* is the boundary thickness correlation constant (mg g^−1^).

The experimental data fitting results are illustrated in Figure 9a,b, while the kinetic parameters are summarized in Table 1. The surface of the AELS initially possessed numerous active sites at the onset of the adsorption process, leading to a rapid increase in adsorption capacity before a more gradual incline, ultimately reaching equilibrium, as illustrated in Figure 9a. Consequently, the active sites on the surface of AELS became saturated over time, causing the adsorption equilibrium to be attained with prolonged adsorption periods. According to the kinetic parameters presented in Table 1, the pseudo-second-order kinetic model provided a more accurate fit to the experimental data compared to the pseudo-first-order and intra-particle diffusion models, as indicated by the higher R^2^ value [37]. This suggested that the adsorption mechanism of methyl blue by AELS predominantly relied on valence forces involving electron exchange or sharing between the AELS and methyl blue molecules [38,39]. Additionally, the Root Mean Square Error (RMSE) associated with the pseudo-second-order kinetics model was lower than that of the pseudo-first-order model, thereby further supporting the conformity of the experimental data to the pseudo-second-order kinetics model.

The diagram in Figure 9b illustrates the relationship between *t*^0.5^ and *q_t_*, displaying the three linear phases of methyl blue as observed in the intra-particle diffusion model. This model encompassed solution diffusion, intra-particle diffusion, and the adsorption of the adsorbent on both internal and external adsorption sites [40,41]. As shown in Table 1, the *k_p_* value exhibited a gradual decrease, indicating a deceleration in the adsorption process. The determination of the *C* value was crucial in assessing whether intra-particle diffusion solely governs the adsorption process. A *C* value of zero signified that intra-particle diffusion was the sole rate-controlling step. However, in this case, the *C* value was not zero, suggesting that external mass transfer also played a role in driving the adsorption process. The increasing *C* value implied a heightened significance of the boundary layer in methyl blue adsorption [42].

#### 3.3.4. Adsorption Isotherm

Experiments were conducted in order to determine the maximum adsorption capacity and adsorption mechanism, at a *C*_0_ range of 10~300 mg L^−1^, *V* = 25 mL, pH = 6, *M* = 0.01 g, *T* = 298 K, 308 K and 318 K. In this study, we employed the Langmuir isotherm model, Freundlich isotherm model and Temkin isotherm model to fit the experimental data [43,44,45]. Equations (7)–(10) of the isotherms were as follows:(7)$$q_{e}=\frac{q_{max}K_{L}C_{e}}{1+K_{L}C_{e}}$$
(8)$$R_{L}=\frac{1}{1+K_{L}C_{0}}$$
(9)$$q_{e}=K_{F}{C_{e}}^{1/n}$$
(10)$$q_{e}=\frac{RT}{b}\ln K_{T}C_{e}$$

In the formulas, *K_L_* (L mg^−1^) is a Langmuir constant related to the maximum adsorption capacity and adsorption energy, *q_max_* (mg g^−1^) is the maximum adsorption capacity of the absorbent, the *R_L_* value represents the feasibility of adsorption, *K_F_* (L mg^−1^) denotes adsorption performance constants, 1/*n* denotes relative adsorption strength parameters related to the Freundlich model, *R* is the universal gas constant (8.314 J K^−1^ mol^−1^), *T* (K) is the thermodynamic temperature, b (J mol^−1^) is heat of adsorption, and *K_T_* (L mg^−1^) is the equilibrium binding constant of Temkin model.

Figure 10 presents the relationship among the adsorption capacity, initial concentration and temperature, along with the fitting results of the experimental data at varying temperatures shown in Figure 10a–c, and relevant isothermal parameters are detailed in Table 2. The adsorption capacity exhibited a significant rise from 24.71 mg g^−1^ to 153.42 mg g^−1^ as the initial concentration escalated from 10 mg L^−1^ to 300 mg L^−1^. This increase in adsorption capacity was attributed to the enhanced interaction between methyl blue and the active sites on the AELS surface, which improved the efficiency of the adsorption process as more methyl blue molecules encapsulate the available active sites. Consequently, the higher initial methyl blue concentration intensified the driving force of the concentration gradient for mass transfer, augmenting the likelihood of collision between methyl blue molecules and AELS, thereby leading to elevated methyl blue uptake [46]. Furthermore, the adsorption capacity increased with the temperature elevation from 298 to 318 K. The accelerated rate of adsorption observed at higher temperatures could be attributed to the heightened mobility of dye molecules, fostering increased interactions between dye molecules and adsorption sites on AELS [47].

The experimental data, as indicated in Table 2, exhibited better agreement with the Langmuir isotherm model, with an R^2^ value of 0.9992. This observation was supported by a comparison of R^2^ values among the Freundlich and Temkin models, which were lower than those of the Langmuir model, signifying the appropriateness of the Langmuir model in fitting the experimental data, thus suggesting that the AELS adsorption of methyl blue involved single-molecular-layer adsorption. Furthermore, the adsorption capacity derived from the experimental data closely approximated the maximum adsorption capacity predicted by the Langmuir adsorption isotherm model. The *R_L_* values, falling within the 0 to 1 range, confirmed the favorability of the adsorption processes across different temperatures [48]. Additionally, the value of *n* was greater than 1, indicating the advantageous nature of the adsorption process [49]. The smaller RMSE values associated with the Langmuir isotherm model compared to those of the Freundlich and Temkin isotherm models indicated a better fit with the Langmuir isotherm model. Moreover, Table 3 presents various adsorption materials for methyl blue, all conforming to the pseudo-second-order kinetic model, with lower maximum adsorption capacities than those of AELS.

#### 3.3.5. Adsorption Thermodynamics

Experiments were conducted at three different temperatures, namely 298 K, 308 K and 318 K, under specified conditions of *C*_0_ = 100 mg L^−1^, *V* = 25 mL, pH = 6 and *M* = 0.01 g. These experiments were analyzed using Formulas (11)–(13) [52] to determine the thermodynamic parameters Δ*G*^0^, Δ*H*^0^ and Δ*S*^0^ associated with the adsorption process:(11)$${∆G}^{0}={∆H}^{0}-T{∆S}^{0}$$
(12)$$\ln K_{d}=\frac{{∆S}^{0}}{R}-\frac{{∆H}^{0}}{RT}$$
(13)$$K_{d}=\frac{C_{0}-C_{e}}{C_{e}}$$

In the formulas, Δ*H*^0^ (kJ mol^−1^) and Δ*S*^0^ (J mol^−1^ K^−1^) are the enthalpy change and entropy change in isothermal adsorption, respectively; Δ*G*^0^ (kJ mol^−1^) is the Gibbs free energy of the adsorption; and *K_d_* denotes adsorption performance constants.

The determinations of the enthalpy change (Δ*H*^0^) and entropy change (Δ*S*^0^) in the adsorption process involved conducting a thermodynamic analysis using the plot of ln*K_d_* versus 1/*T*. The fitting result of the thermodynamic is displayed in Figure 11, and the linear equation parameters are shown in Table 4. According to Table 4, the Δ*G*^0^ obtained from the adsorption process at different temperatures was negative, suggesting that the methyl blue adsorption on the AELS was spontaneous. The adsorption process was thermodynamically stable with increasing temperatures from 298 K to 318 K, as the value of Δ*G*^0^ became more negative with the increase in temperature, indicating that the adsorption process was more favorable at higher temperatures [10]. The positive Δ*H*^0^ value revealed that temperature elevation enhanced adsorption due to its endothermic nature [53]. Furthermore, Δ*S*^0^ > 0 implied that the degree of freedom at the material solution interface increased during the adsorption process. Consequently, the thermodynamic experiments empirically confirmed that the adsorption of methyl blue onto AELS was not only spontaneous but also an endothermic process.

#### 3.3.6. Regeneration and Reusability of AELS

To understand the regeneration and reusability of the adsorbent, the saturation of the adsorbent was carried out at *C*_0_ = 50 mg L^−1^, *V* = 25 mL, pH = 6, *M* = 0.01 g and *T* = 298 K. The adsorbent saturated with the target pollutant underwent a series of elution steps by mixing it with 20 mL of eluent (0.1 mol L^−1^ HCl) and sonicating the mixture several times. Subsequently, the adsorbent was meticulously rinsed with deionized water in preparation for its reuse in the subsequent adsorption cycle. Notably, the difference observed between cycles was deemed insignificant. The result is shown in Figure 12. After five cycles, it demonstrated a decrease in adsorption capacity from 118.99 mg g^−1^ to 114.33 mg g^−1^, underscoring the commendable regeneration and reutilization potential of the AELS. These findings underscore the efficiency of AELS as a viable adsorbent for methyl blue removal in dye wastewater applications, attributing its success to both its economic viability and reusability.

#### 3.3.7. Adsorption Mechanism

The analysis of the AELS characterization revealed that AELS had a porous, blocky structure with numerous functional groups present on the surface. Methyl blue, a molecule with a complex structure containing various functional groups like -SO_3_^−^, =N-, phenyl- and -NH, would likely interact specifically with the functional groups of the dye molecules on the adsorbents. There are various potential interactions between dye molecules and adsorbents. Research on the adsorption kinetics and isotherms demonstrated that the adsorption mechanism of methyl blue on AELS included significant electrostatic interactions and hydrogen bonding. The sulfonic acid groups could undergo electrostatic interactions with positively charged AELSs; the -OH group on the surface of the AELS forms hydrogen bonds with the N-H groups in the methyl blue molecule. A schematic of the adsorption mechanism is shown in Figure 13.

## 4. Conclusions

In this research, AELS was successfully synthesized through the Mannich reaction and demonstrated as an efficient adsorbent for removing methyl blue from aqueous solutions. The characterization of the synthesized material was conducted using FT-IR, TG, SEM and XPS analyses. Various factors, including pH, adsorbent dosage, contact time, initial concentration and temperature, were found to impact the adsorption capacity of AELS for methyl blue. It was observed that the adsorption capacity decreased with an increasing pH and adsorbent dosage. The adsorption kinetics and isotherms were effectively described through the pseudo-second-order kinetic model and Langmuir isotherm model, respectively. The adsorption process was single-molecular-layer and the maximum adsorption capacity of AELS for methyl blue was determined to be 153.42 mg g^−1^. Thermodynamic studies revealed that the adsorption process was spontaneous and endothermic, driven by strong electrostatic interactions and hydrogen bonding. The adsorption capacity remained consistent after five cycles, indicating the potential for the reusability of AELS. This study not only presents a novel approach for utilizing lignin, but also offers an efficient and cost-effective method for treating dye wastewater through adsorption.

## Figures and Tables

**Figure 1 materials-17-01046-f001:**
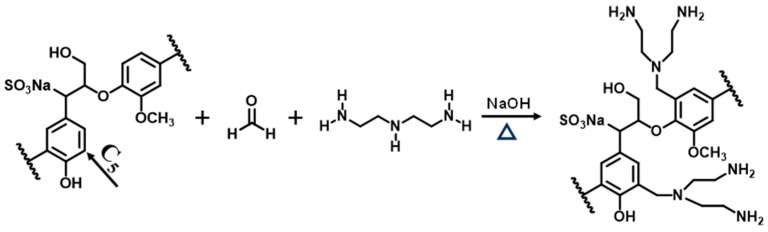
The schematic representation of the synthesis of AELS.

**Figure 2 materials-17-01046-f002:**
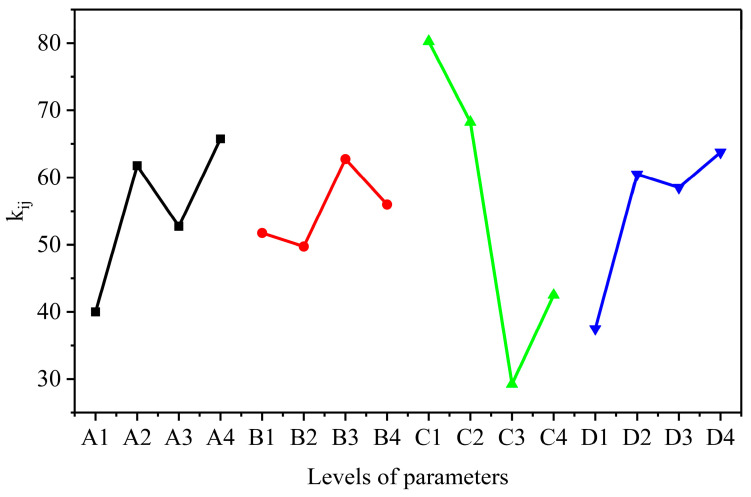
The effects of synthesis conditions on adsorption capacities: reaction temperature (A, black), reaction time (B, red), DETA quantity (C, green) and FA quantity (D, blue).

**Figure 3 materials-17-01046-f003:**
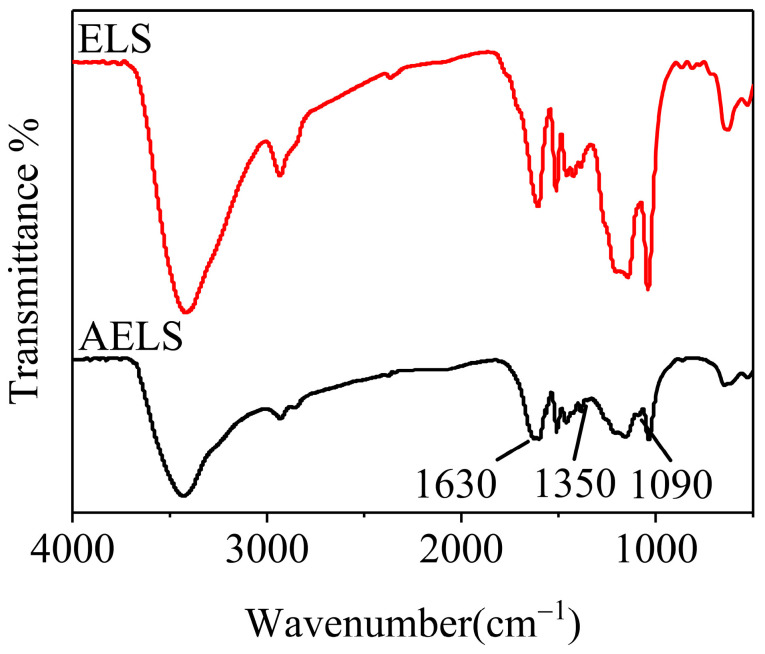
FT-IR images of ELS and AELS.

**Figure 4 materials-17-01046-f004:**
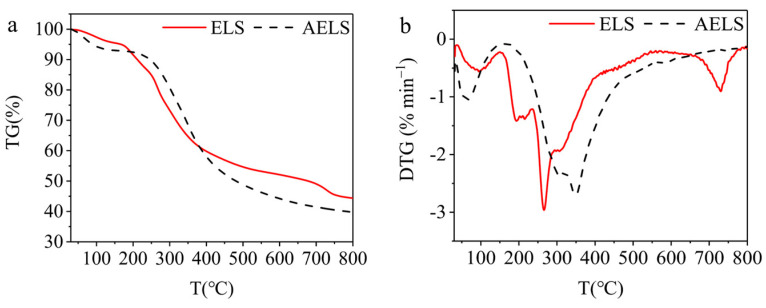
TG and DTG images of ELS and AELS: (**a**) TG images; (**b**) DTG images.

**Figure 5 materials-17-01046-f005:**
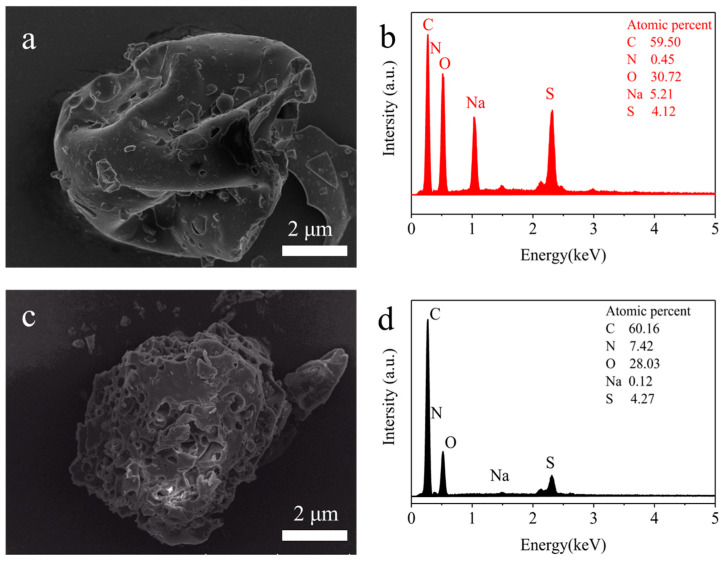
SEM and EDS images of samples: (**a**) SEM images of ELS; (**b**) EDS images of ELS; (**c**) SEM images of AELS; (**d**) EDS images of AELS.

**Figure 6 materials-17-01046-f006:**
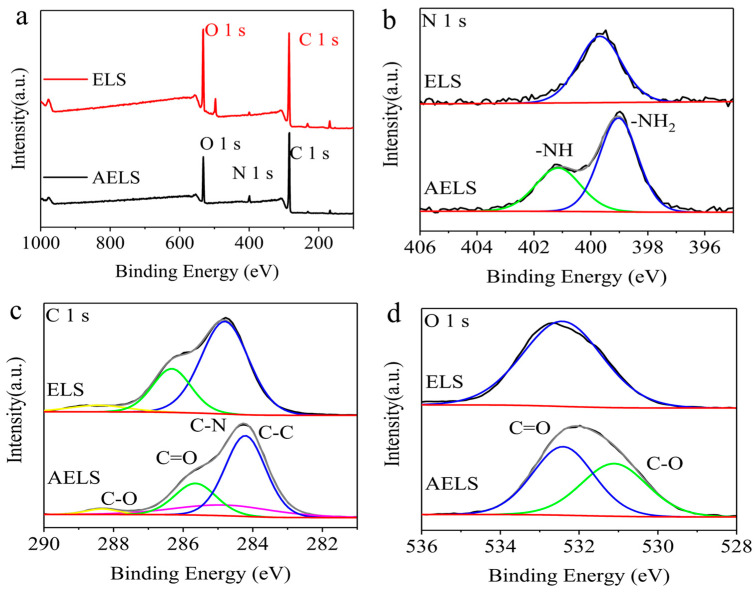
XPS patterns of ELS and AELS. (**a**) Full spectrum of sample; (**b**) N 1 s XPS spectra of sample; (**c**) C 1 s XPS spectra of sample; (**d**) O 1 s XPS spectra of sample.

**Figure 7 materials-17-01046-f007:**
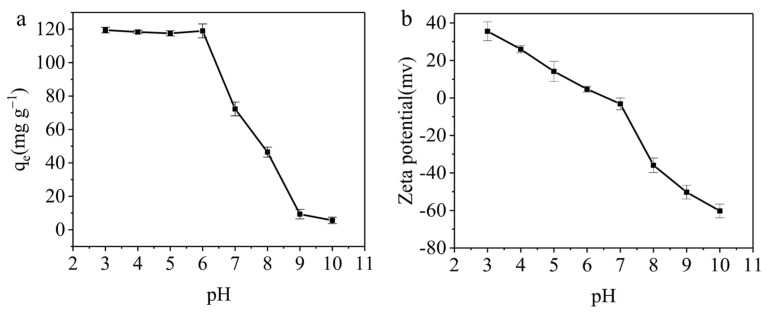
The effect of pH: (**a**) The effect of pH on the adsorption capacity. (**b**) Zeta potential of AELS.

**Figure 8 materials-17-01046-f008:**
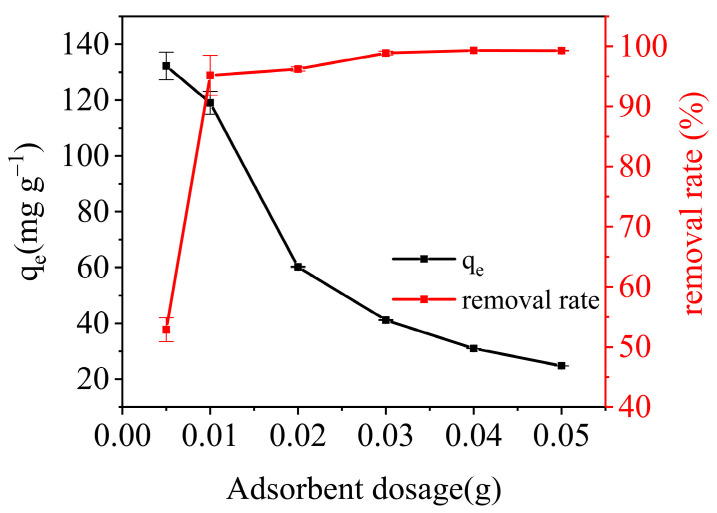
The effect of adsorbent dosage on the adsorption capacity and removal rate.

**Figure 9 materials-17-01046-f009:**
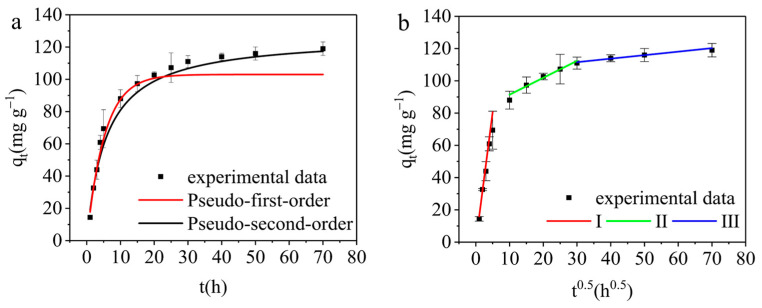
Kinetic analysis of AELS adsorption for methyl blue: (**a**) Experiment data, pseudo-first-order kinetic model fitting and pseudo-second-order kinetic model fitting; (**b**) intra-particle diffusion kinetic model fitting.

**Figure 10 materials-17-01046-f010:**
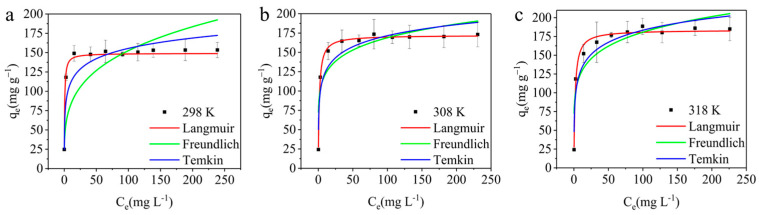
Isotherm analysis of AELS adsorption for methyl blue: (**a**) 298 K; (**b**) 308 K; (**c**) 318 K.

**Figure 11 materials-17-01046-f011:**
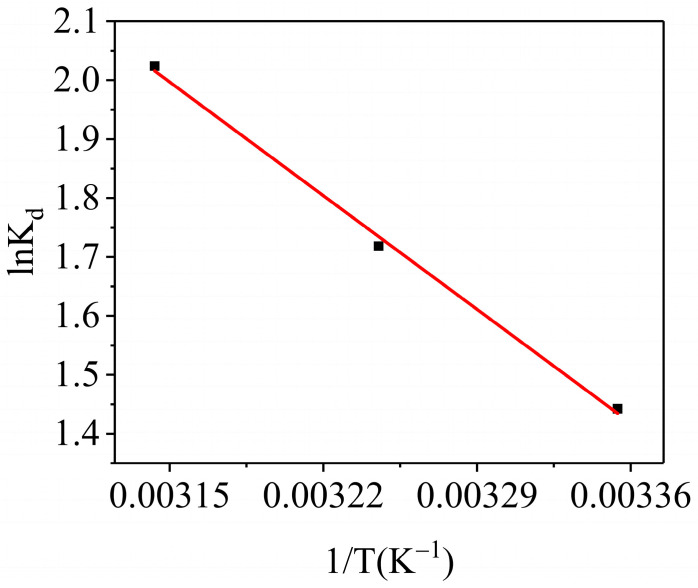
Thermodynamic fitting diagram.

**Figure 12 materials-17-01046-f012:**
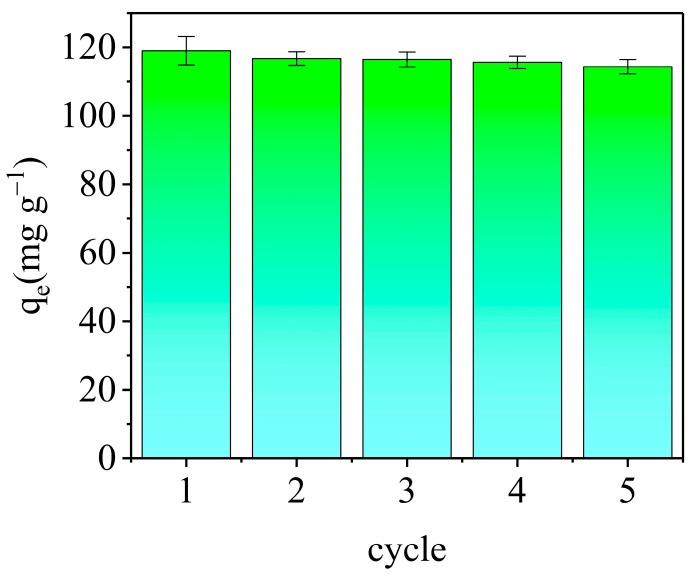
The regeneration and reusability of AELS.

**Figure 13 materials-17-01046-f013:**
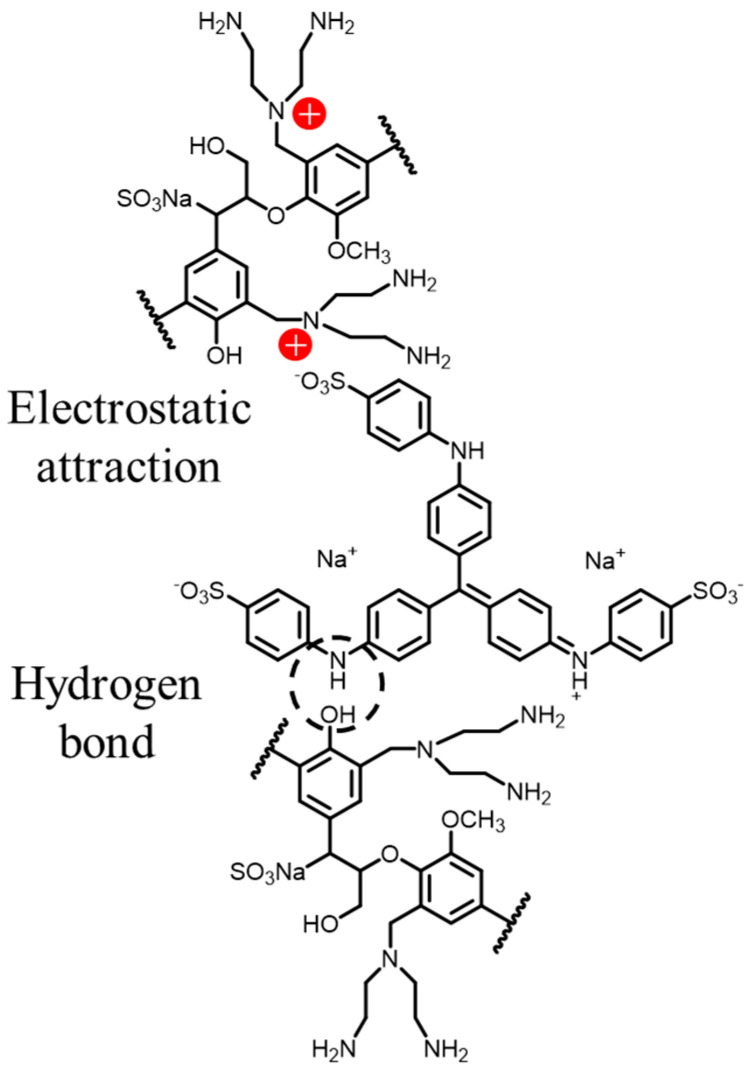
Schematic of adsorption mechanism.

**Table 1 materials-17-01046-t001:** Pseudo-first-order kinetic, pseudo-second-order kinetic and intra-particle diffusion model fitting parameters.

Dynamic Model	Pseudo-First-Order Kinetics		Pseudo-Second-Order Kinetics		Intra-Particle Diffusion			
						I	II	III
	R^2^	0.9852	R^2^	0.9930	R^2^	0.9754	0.9324	0.9279
Parameters	*k* _1_	0.19 ± 0.0085	*k* _2_	0.0014 ± 0.0004	*k_p_*	16.17 ± 1.28	1.05 ± 0.14	0.22 ± 0.034
	*q_e_* RMSE	103.00 ± 0.004 2.03	*q_e_* RMSE	126.94 ± 0.003 1.39	*C*	−0.22 ± 2.51	80.94 ± 2.98	105.06 ± 1.43

**Table 2 materials-17-01046-t002:** Langmuir, Freundlich and Temkin isotherm model fitting parameters.

T/K	298	308	318
	Langmuir		
*q_max_* (mg g^−1^)	149.08 ± 1.30	172.14 ± 1.23	183.69 ± 2.67
*K_L_* (L mg^−1^)	1.74 ± 0.022	0.69 ± 0.050	0.59 ± 0.08
*R_L_*	0.0019–0.054	0.005–0.13	0.006–0.15
R^2^	0.9992	0.9961	0.9850
RMSE	0.51	2.77	5.93
	Freundlich		
*K_F_* (L mg^−1^)	44.66 ± 2.091	87.21 ± 13.88	87.61 ± 13.42
*n*	3.75 ± 0.22	6.96 ± 1.74	6.36 ± 1.40
R^2^	0.9194	0.7823	0.8240
RMSE	5.04	20.62	20.28
	Temkin		
*b* (J mol^−1^)	128.46 ± 7.06	120.20 ± 15.39	106.36 ± 11.15
*K_T_* (L mg^−1^)	31.69 ± 2.66	41.24 ± 39.30	26.21 ± 19.14
R^2^	0.9766	0.8850	0.9199
RMSE	2.71	14.98	13.68

**Table 3 materials-17-01046-t003:** Methyl blue-adsorbing materials and their related parameters.

Adsorbent	Kinetic Model	Isotherm Model	*q_max_* (mg g^−1^)	Reference
Mn/MCM-41	Pseudo-second-order	Freundlich Dubinin–Radushkevich	45.38	[14]
KGM/GO hydrogel	Pseudo-second-order	Freundlich	133.67	[15]
Organo bentonite	Pseudo-second-order	Freundlich	98.15	[16]
Magnetic NiFe_2_O_4_ nanorods	Pseudo-second-order	Temkin	97.73	[50]
Mg_3_Al LDH	Pseudo-second-order	Langmuir	98.77	[51]
AELS	Pseudo-second-order	Langmuir	153.42	This work

**Table 4 materials-17-01046-t004:** Thermodynamic fitting parameters.

Thermodynamic Parameter	Δ*H*^0^ (kJ mol^−1^)	Δ*S*^0^ (J mol^−1^ K^−1^)	Δ*G*^0^ (kJ mol^−1^)		
			298 K	308 K	318 K
	22.91	88.76	−3.55	−4.44	−5.33

## Data Availability

Data are contained within the article and Appendix A.

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
