# Peer review of "Preparation of Aminated Sodium Lignosulfonate and Efficient Adsorption of Methyl Blue Dye"

_materials, 2024, doi:10.3390/ma17051046_

Round 1
Reviewer 1 Report
Comments and Suggestions for Authors
Journal: Materials
Manuscript ID: materials-2852884
Title: Preparation of aminated sodium lignosulfonate and efficient adsorption of methyl blue dye
Huo et al. aim to address water pollution resulting from dye wastewater by focusing on the preparation and optimization of aminated sodium lignosulfonate (AELS) through the Mannich reaction. The study aims to determine the optimal conditions for amination through orthogonal experiments and subsequently validates the successful synthesis of AELS using various characterization techniques such as FT-IR, XRD, TG, SEM, and XPS. The primary objective is to investigate the efficiency of AELS as an adsorbent for Methyl blue (MB) dye under various conditions, including pH, adsorbent dose, initial concentration, time, and temperature. Furthermore, the manuscript aims to explore and discuss the kinetics, isotherms, and thermodynamic properties of the AELS adsorption for MB, ultimately elucidating the underlying mechanisms. The research seeks to contribute to the effective utilization of natural resources, specifically lignin, promoting sustainability and uncovering new applications for lignin in the context of water pollution remediation. The topic is interesting and fits well with the scope of the journal. The results are promising, and the manuscript is significantly improved. Still, some issues remained unaddressed. My specific comments are given bellow.
-I still think the authors should use non-linear equations for experimental data fitting (kinetic models and isotherms).
-The abstract should contain some values, at least for adsorption capacity.
-The authors should suggest the mechanism of adsorption.
Comments on the Quality of English Language
Minor changes are required.
Reviewer 2 Report
Comments and Suggestions for Authors
There is no doubt about the relevance of adsorption as a method to clean wastewater. From this point of view, the manuscript is relevant, although it should be significantly improved. I hope that the authors can consider my comments.
The text should be reviewed again for general writing, such as using capital letters where they should be in lowercase.
Line 77
What do AR and BS mean?
The material and methods section does not specify whether the authors used controls, the number of repetitions of the experiments, and the statistical treatment the data received. Besides, figures about sorption experiments do not show error bars, and numerical results such as sorption parameters (in the text and tables) do not present the standard deviations.
Lines 203-205
The authors found little difference in the adsorption capacity between pH 3 and 6. Statistically, is that little significant difference or not (p=???)?
Moreover, they argued the pH of the MB solution did not change in subsequent experiments. But which pH was selected for further experiments?
Section 3.2.3
In section 2.4, the authors specified that the experiments were carried out at a "certain adsorbent dosage, a certain MG concentration, and a constant temperature", but in section 3.2.3, none of these conditions appear. The pH solution is also not specified.
Table 2.
I suggest to the authors a change in Table 2. It is better after the isotherm model fitting (currently Table 3).
Lines 278-279
The manuscript specifies that the high concentration gradient facilitated MB transportation and improved the resistance barrier between solid and aqueous phases.
The phrase "improve the resistance barrier". Does it mean that resistance increases or diminishes?
Figure 9a
The abscissa axis is Ce or only C?
Figures 8c and 9b
Units of the ordinate axes.
3.2.4. Adsorption isotherm section.
Finally, the authors did not compare the maximum biosorption capacity of AELS with other adsorbents.
Adsorption thermodynamics.
Does the value of ΔH have any possible explanation?
Do the sign and value of ΔS have any explanation?
3.2.6. Regeneration and reusability of AELS
The authors did not explain this part of the study in the materials and methods section. Reporting the experimental conditions is essential, and the eluent they used is of great interest.
Likewise, figure 11 does not show the errors between the bars, and it is not clarified in the text whether the differences between the cycles are significant.
Lines 340-341
"The adsorption capacity was essentially unchanged after five cycles". Was it determined according to the statistics?
Reviewer 3 Report
Comments and Suggestions for Authors The manuscript cannot be published in this form and requires major revisions.See the detailed review report.

See the detailed review report.
Reviewer 4 Report
Comments and Suggestions for Authors
1. Extend the chapter Conclusions.
2. Add the chemical formula of methyl blue.
3. Not all abbreviations are explained (e.g. BS, AR,).
4. Fig. 10 The fiitting diagram has only 3 experimental points. The scientific meaning is very low.
5. How was the adsobent regenerated? Chapter 3.2.6 needs more explanation.
Comments on the Quality of English LanguageNo comment.
Round 2
Reviewer 2 Report
Comments and Suggestions for Authors
I thank the authors for responding favorably to my previous suggestions.
In the new version, I only want to point out some details.
Table 3
What are the units of qmax? mg/g?
3.2.6. Regeneration and reusability...
The authors mention:
"To understand the regeneration and reusability of adsorbent, experiments were carried out at an initial concentration of 50 mg L-1, the volume of 25 mL, pH of 6, adsorbent dosage of 0.01 g, and temperature of 298 K."
I understand that instead of "experiments were carried out..", it should be "the saturation of the adsorbent was carried out at an initial concentration of 50 mg L-1 (of Methyl blue)..."?
Reviewer 3 Report
Comments and Suggestions for Authors
The article has been greatly improved and can be published.
Comments on the Quality of English Language
Minor editing of English language required
